# Automatic Recognition of Oil Spills Using Neural Networks and Classic Image Processing

**Rotem Rousso [1], Neta Katz [1], Gull Sharon [2], Yehuda Glizerin [2], Eitan Kosman [2] and Assaf Shuster [2,*]**

1    Electrical and Computers Engineering Department, Technion—Israel Institute of Technology, Haifa 32000, Israel; rotem.rousso@campus.technion.ac.il (R.R.); neta-katz@campus.technion.ac.il (N.K.)

2    Computer Science Department, Technion—Israel Institute of Technology, Haifa 32000, Israel; gull.sharon@campus.technion.ac.il (G.S.); gyehuda@campus.technion.ac.il (Y.G.); eitan.k@cs.technion.ac.il (E.K.)

\*    Correspondence: assaf@technion.ac.il

**Abstract:** Oil spill events are one of the major risks to marine and coastal ecosystems and, therefore, early detection is crucial for minimizing environmental contamination. Oil spill events have a unique appearance in satellite images created by Synthetic Aperture Radar (SAR) technology, because they are byproducts of the oil's influence on the surface capillary, causing short gravity waves that change the radar's backscatter intensity and result in unique dark formations in the SAR images. This signature's appearance can be utilized to monitor and automatically detect oil spills in SAR images. Although SAR sensors capture these dark formations, which are likely connected to oil spills, it is hard to distinguish them from ships, ocean, land, and other oil-like formations. Most of the approaches for automatic detection and classification of oil spill events employ semantic segmentation with convolutional neural networks (CNNs), using a custom-made dataset. However, these approaches struggle to distinguish between oil spills and spots that resemble them. Therefore, developing a tailor-made sequence of methods for the oil spill recognition challenge is an essential need, and should include examination and choice of the most effective preprocessing tools, CNN models, and datasets that are specifically effective for the oil spill detection challenge. This paper suggests a new sequence of methods for accurate oil spill detection. First, a SAR image filtering technique was used for emphasizing the unique physical characteristics and appearance of oil spills. Each filter's impact on leading CNN architectures performances was examined. Then, a method of a model ensemble was used, aiming to reduce the generalization error. All experiments demonstrated in this paper confirm that using the sequence suggested, in comparison to the common formula, leads to a 4.2% of improvement in the intersection over union score (IoU) for oil spill detection, and a 9.3% of improvement in the mean IoU among several relevant classes.

**Keywords:** convolutional neural network; image filtering; pre-processing; oil spill detection; SAR imagery; semantic image segmentation; ensemble modeling; marine oil spill

## 1. Introduction

Oil spills are a serious concern for the ecosystem of shorelines and reefs due to the damage they cause. They can severely harm the coastal ecology, causing water pollution, which takes effort to clean, and poisoning marine life [1], which may have an indirect effect on habitats in which plants and animals live [2]. Moreover, marine oil spills have major economic and social impacts, such as imposing costs and negative impacts on fishing and marine industries [3]. The oil spill phenomenon is widespread; according to [4], between 1970 and 2010, approximately 5.71 million tons of oil were leaked from ships. Thus, there is a major international focus on finding solutions for detecting and minimizing these events, and especially identifying methods of minimizing response time and enhancing the detection accuracy of oil spill recognition. The main challenge is to

overcome the misclassification of similar-looking dark spots using Synthetic Aperture Radar (SAR) images.

SAR sensors are mainly used to monitor and capture the marine status and events. SAR is a microwave-based technology that emits radio waves and receives their reflection, thereby capturing a representation of the target scene, widely referred to as SAR images [1]. Sentinel-1A, launched by the European Space Agency (ESA), is a widespread source for images used in marine remote sensing systems. Two identical SAR satellites with programmable polarizations continuously provide data on oceans, land changes, ships, and oil. The data from the Sentinel-1 SAR repository is free and available for registered users through ESA's Sentinel-1 Internet data hub. This dataset was chosen for examination and comparison of our method's performances since it has been used in both research studies and operational use for detecting ocean surface events [5]. However, there are still challenges in using SAR images to detect oil spills despite the vast amount of data SAR provides.

One of the main challenges in oil spills detection is the high frequency of false oil detections, referred to as 'oil look-alikes'. Both oil spills and oil look-alikes appear as dark features in the SAR image, unlike more bright water and other ocean or land features. Oil spills influence the surface capillary, causing short gravity waves, which change the radar's backscatter intensity and thus creating dark formations [6]. Although the appearance of these dark oil spills features is rare, similar dark features also appear in the SAR images, such as grease ice, current shears, internal waves, wind shelters, and unknown objects [3]. To date, research on this challenge has been handled by creating a new class, known as Oil Look-Alike, so any non-oil spill dark spot is labeled as Oil Look-Alike. One of the consequent method challenges is developing a semantic segmentation system that can distinguish between oil spills and oil look-alikes.

The content used for oil spill detection captured by SAR sensors also contains ships, ocean, and land appearances, which have been found to be elements that can indicate the likelihood of an oil spill event in their area. Thus, by designing an automatic detection model that can classify them, the overall detection of oil spills may also be improved. A neural network that can help the early detection of oil spills, both in a specific region and in general, can alert relevant authorities earlier and hasten the response to such disasters. As part of the suggested detection sequence, this research proposes using deep neural network learning models aimed at analyzing SAR image datasets, which can help in decision making via the semantic segmentation of ocean oil spill classification.

Some previous results have shown that using convolutional neural networks (CNNs) outperformed classic signal processing techniques used solely in many tasks and applications [7]. CNNs can be trained in an end-to-end fashion for mapping an input image to the desired output. This knowledge can be generalized to classify new unseen SAR image segmentations. By leveraging the ability of CNNs to be re-trained using a dataset specified for the oil spill detection challenge, a new state-of-the-art method to detect oil spills may be achieved, compared to the classic methods of pattern recognition [8], which tend to be more domain specific [9]. We believe that in addition to using only CNNs, combining them with classic image processing techniques can lead to better and more accurate performances than the current state-of-the art techniques. For comparison and performance evaluation, our methods were applied to extensive experiments on the Oil Spill Detection Dataset, which was also used in a 2019 study by Krestenitis et al. [10]. This dataset is split into 1002 images designated for training, and another 110 images designated for testing the evaluation and for comparison.

The CNN training process requires a large and varied dataset of images for oil spill automatic recognition but, in its original form, the SAR dataset we examined does not contain enough images. Therefore, we implemented image augmentation methods on the training dataset [11]. In data analysis, data augmentation refers to strategies that increase the quantity of data by adding slightly changed copies of current data or creating new

synthetic data from existing data. When training a machine learning model, these added samples work as a regulator to alleviate overfitting [12].

Oil spills' special appearance has unique physical characteristics on SAR images. Attempting to differentiate oil spills from oil look-alike backscatter intensity, methods that enhance and distinguish oil spills' characteristics from other phenomena were searched for. As a result, we found that adding traditional image filtering as a pre-processing step outperforms CNNs' baseline form.

In a previous work, an attempt to include a speckle filter followed by a median filter was examined [10], without intensifying the disparities between the labels. Another study [13] has suggested using image filtering prior to the CNN step, while using different datasets, by applying the Refined Lee Filter for noise reduction. A study also tested threshold segmentation techniques on the results [14], which enhanced the appearance of oil spills and oil look-alikes while resulting in the loss of valuable information from the original images

Although there have been several attempts to use the combination of both classic image filtering methods and CNNs for accurate results in the oil spill detection challenge, many of these have caused information loss, while look-alike misclassifications have remained a problem. We hypothesized that the classification of actual oil would be improved by focusing on examining filters and methods based on specifically emphasizing the physical unique characteristics of oil spills in SAR images, that would differentiate them from oil look-alikes, without causing a significant information loss. Since this theory has not been observed yet, we used a variety of image filters on the dataset, including histogram equalization and contrast stretching. The filters used overcome both information loss and misclassification challenges.

Our goal was to create a systematic sequence of methods, including the use of CNN architectures for the semantic segmentation step, and use several classic image processing methods adjusted for each label's most accurate detection, all combined with an ensemble model and applied to the dataset. Specifically, the application of image filters to the dataset was examined, such as histogram equalization and contrast stretching, alongside the implementation of augmentations on the training dataset. These methods were implemented to determine the most beneficial solutions compared to existing detection methods, and their effectiveness was measured. Using this combination of selected techniques resulted in a decrease in the oil spill vs. look-alike false detection rate while increasing the mean score by 9.3% in terms of the intersection over union (IoU), compared to the current state-of-the-art approaches [10].

The current paper is organized as follows: Section 2 describes existing methods that led to the improved accuracy in classification identification. The CNN architectures that were used, Unet and DeepLabv3+, are described and the motivation for using them. This section also explains the data pre-processing methods that were applied; how and why the data augmentation was implemented; the image filtering; and how the model ensemble impacted detection accuracy. Section 3 discusses the numerical results and performance rates, and compares them to the current state-of-the-art approaches in this field. Section 4 summarizes the conclusions from our research and the implementation process, and presents suggestions for future work.

## 2. Methods

This section details the discussion of the methods we used to form the semantic segmentation system that assists in highlighting the differences between the labels. The system design concept is presented, in addition to the theoretical research and development of the system. This section also details the dataset used, the pre-processing tools, and the architectures examined.

*2.1. CNN Architectures*

The main goal was to classify each pixel in a SAR image into one of the following classes: Ocean, Oil Spill, Oil Look-Alike, Ship, and Land. This required an analysis of all the pixels in the input image. Each image used a 2-dimensional array of integers in the range 0 to 255, where every integer indicates the grayscale value of a pixel coming from a processed SAR image. The main analysis method applies a CNN whose output would be one of the classes mentioned above, tagged for each pixel of the input SAR image. The CNN algorithm takes multidimensional arrays known as tensors for input; these tensors are sequentially passed through different layers of processing that may use convolution, pooling, normalization, full connection, and activation. The processed 2-D array SAR image was used as the input tensor. To segment the pixels into classes as output, the SoftMax activation function was used as the final layer for the CNNs. SoftMax was chosen due to its ability to normalize the network's output to a probability distribution over the predicted classes: Ocean, Oil Spill, Oil Look-Alike, Ship, and Land. Equation (1) expresses the normalized output given by the function:

$$\sigma(z)_i = \frac{e^{z_i}}{\sum_{j=1}^{K} e^{z_j}}, \forall i \in \{1, 2, 3, 4, 5\} \tag{1}$$

where $z_1, z_2, z_3, z_4, z_5$ are the original prediction for the classes and $\sigma(z)_1$, $\sigma(z)_2$, $\sigma(z)_3, \sigma(z)_4$, $\sigma(z)_5$ are the respective predictions normalized by the SoftMax function. Based on the SoftMax outputs, we create a predicted segmentation map for each pixel by choosing the class that achieved the highest SoftMax score; this can be interpreted as the class in which the model predicts that the pixel will most likely be classified. Further details of the architectures used are presented in the following subsections.

2.1.1. Unet

Unet is a CNN architecture for image segmentation initially proposed to deal with semantically segmenting biomedical images. The Unet model contains two parts: the contracting path (i.e., the encoder) and the expansive path (i.e., the decoder).

The encoder incrementally decreases the tensor it receives while increasing the number of features that it contains, to capture the meaningful content of the image into a dense vector of features. Various implementations are offered for the encoder; typically known as backbones, they are often chosen to be other CNNs that are trained for classification, sharing the same structure that incrementally decreases the size of the tensor and increases the number of features. Originally, the implementation of the encoder consisted of four layers, where each layer consists of two $3 \times 3$ convolutional layers, followed by a $2 \times 2$ max pooling operation with stride 2.

After the encoder incrementally compresses the tensor, the decoder gradually upscales the tensor back to its original size, mapping the meaningful features to their respective locations. At every step, the tensor is initially up-sampled to increase the size of the tensor. Then, it has a $2 \times 2$ convolution performed on it for decreasing the number of features. Following that, it is concatenated to the tensor that was at the matching level of the encoder for avoiding the loss of information. Then, two consecutive $3 \times 3$ convolutions are performed, further decreasing the number of features. The final step of the decoder is a $1 \times 1$ convolution to change the feature map for matching the number of output classes.

The ResNet-101 architecture was used for the encoder since it demonstrated good performance in previous classification studies on the ImageNet dataset, which was designed for visual object recognition software [15]. In addition, ResNet-101 was the backbone used in the baseline paper for the Unet model [10].

2.1.2. DeepLabv3+

DeepLabv3+, which is an upgrade of the DeepLab architecture used for segmentation, was also examined.

DeepLabv3+ combines two important abilities: (1) it probes the incoming features with filters or pooling operations at several rates and multiple effective fields-of-view to encode multi-scale contextual information; and (2) it captures sharper object boundaries by gradually recovering the spatial information [16]. As opposed to its predecessor, DeepLabv3+ has an effective decoder that refines the segmentation results and can produce distinctive object boundaries [10]. The decoder up-samples the outcome of the main branch encoder. As input, it also uses the low-level feature map that was extracted from the encoder backbone and processed by a convolutional layer. All inputs are concatenated into a feature map containing another two convolutional layers. The output is bi-linearly up-sampled, so the original dimensions of the given image are recovered. For more details about DeepLabv3+, the reader is referred to [16].

The MobileNetV2 architecture for the encoder was chosen since it has fewer parameters in the architecture, enabling efficient and fast learning for the model. In addition, MobileNetV2 was the backbone used in the baseline paper for the DeepLabV3+ model [10].

*2.2. Data Pre-Processing*

2.2.1. Sentinel-1 Dataset

In order to learn the characteristics of the different classes, the CNN system requires a large database of satellite images.

Research on oil spill detection lacked a common database of images until a 2019 study by Krestenitis et al. [10], which succeeded in providing a well-established dataset that can be used to identify oil spills by analyzing SAR photos. The dataset also includes semantically annotated masks that enable researchers to evaluate their experimental results. This research used a dataset created by the European Space Agency (ESA), which gathered the SAR images, and the European Maritime Safety Agency (EMSA), both providing information on the geographic coordinates and timestamps via the Clean Sea Net service. Oil pollution records from 28 September 2015 to 31 October 2017 were used, and the SAR images are from the European Sentinel-1 satellite missions.

The dataset described was used both for training and evaluating in this research. More details on this dataset are available in Krestenitis et al., 2019 [10] (p. 4).

2.2.2. Data Augmentation

The original dataset used contains 1112 varied SAR images, taken by Sentinel-1 and supplied after being pre-filtered with a speckle filter followed by a $7 \times 7$ median filter [10].

The CNN learning method requires a large and varied dataset, meaning that our CNN input would have to include a varied set of oil spill instances. Having only about 1000 images was not enough to reach high accuracy for oil spill detection, so a method to augment the dataset was needed.

One of the challenges faced by automatic recognition is preventing overfitting in the learning process of CNN. Another challenge is generalizing the oil spill instances in the input images. This can be achieved by applying image augmentation methods during the training phase for each epoch and directly on the original dataset before the CNN algorithm is applied.

Data was augmented using the following random transformations: The first transform applied was a random image horizontal and vertical flip, which reverses the pixels along the rows or columns and mirrors the image horizontally and vertically. Using these mirroring transforms ensures that there are no dependencies between rows or columns in the learning process. This allows the labeling step to retain generalization per pixel. The second transform applied was image rotation, which rotates the image within affine transformations at a random angle for each epoch. In this way, the oil spill characteristics are generalized for every capturing angle needed to be labeled in the detection output. The image perspective transformation, which converts a 3-dimensional image into a 2-dimensional one, and vice versa, was also used. This prevented the recognition of each class of capturing the image.

To avoid overfitting and for making sure that the augmentations remained varied from one epoch to another, all the augmentations mentioned above were applied using random combinations. The type of augmentation was chosen randomly. A binary threshold of 0.5 from the uniform range of 0 to 1 probability was used for determining whether each augmentation type for every composition should be included. When image rotation was applied, the degree of rotation was also chosen randomly from the range of 90 to 270 degrees. When image perspective was applied, its distortion scale was set to 0.6. for each epoch, such that each epoch would have a different composition of transformation applied to the images.

The same input modifications to the segmentation masks for every pixel were applied, ensuring that the input segmentation masks matched the image and that the target segmentation masks remained after the transformations were applied (e.g., Figure 1). The size of the image and the relative location of each pixel to its neighboring pixels are not necessarily preserved after the transformation. Therefore, interpolation of some pixel values in the augmented image included ensuring the same interpolation method was applied for both the image and its labeled mask.

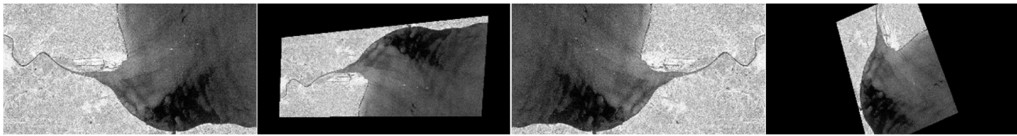

**Figure 1.** Different augmentations applied to the same image. From left to right: original image; image perspective with horizontal flip composition; vertical flip augmentation only; horizontal, vertical, and image rotation combined.

Previous studies mostly used the default bi-linear filter as the interpolation method, but we chose an interpolation method based on the nearest neighbor approach [17].

In the dataset labeling method, every class is represented by a number. Therefore, to maintain correspondence between image, labels, and mask augmentations, the bi-linear filter interpolates the label's class for the artificial pixels added in the perspective transformation. Every label for each class is represented by a different number. When using the bi-linear filter, the label for the interpolated pixels is set by the weighted average of the surrounding neighbor pixels' numerical representation. As a result, the average value of each label does not necessarily represent an existing class. It can even represent an existing incorrect class that does not relate to the value of the edge labels. In short, the bi-linear filter causes every mask to have an incorrect (average, by number) labeling for each edge pixel between 2 classes as depicted in Figure 2, which leads to undesirable misclassification.

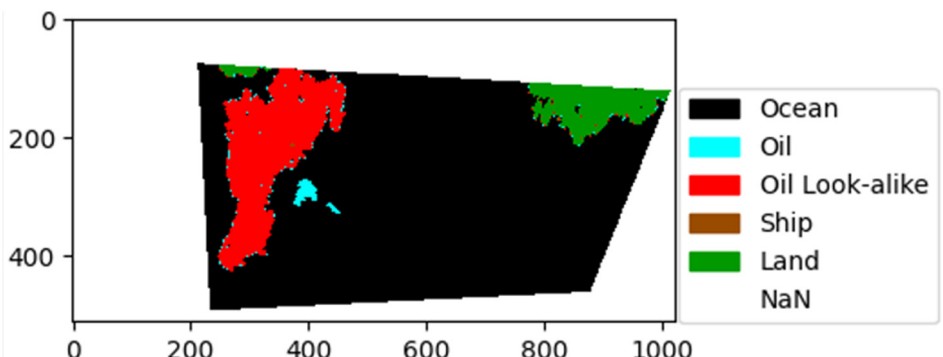

**Figure 2.** Incorrect labeling (Oil Spill) around edges in the outlines of the Oil Look-Alike area. Bi-linear default interpolation—mistaken label interpolation caused after augmentation.

Using the nearest neighbor interpolation method prevents this issue by labeling the new pixels of the augmented image based on the prevalence of existing labels in the

neighboring pixels (Figure 3). Thus, we chose to use this option of interpolation on the original image and its labeled mask.

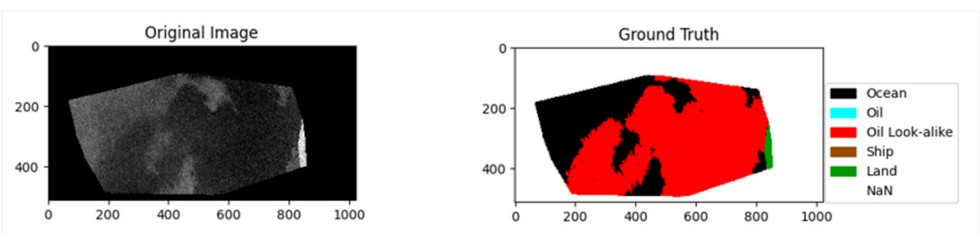

**Figure 3.** Correct labeling around edges. Nearest neighbor interpolation—label interpolation after augmentation is applied correctly.

*2.3. Image Filtering*

Image filtering is a technique for altering the size, colors, outlines, shading, and other characteristics of an image. Each filter affects different features of the image characteristics. Several types of filters were applied to the image to find those that emphasize the image's edges or outline, and better influence the detection performance. Ultimately, the goal was to find methods that emphasized the differences between each class in the original image.

These filters are used in transforming the images and are based primarily on mathematical convolutions between the image and a specific kernel that is used to create the desired effect upon the image.

We set out to find the optimal image filter that would help the CNNs differentiate between one class and another. It was found that there is a unique set of filters that achieved the best accuracy for each class. In this research, the effect of applying histogram equalization and contrast stretch filters to the SAR images was examined, prior to CNN classification.

The histogram equalization filter balances the grayscales of every SAR image, based on the image probability distribution. The general idea behind this filter is to re-assign the pixels' intensity values to make intensity distribution as uniform as possible. This is a simple method that enhances a low contrast image when there is no apparent contrast change between an object and its background [18]. In our case, the important objects to detect were Oil Spill and Oil Look-Alike, located in the ocean. Consequently, the filter would ideally enhance the contrast in these regions and highlight the differences between these classes based on their texture, which also indicates that this filter also created some noise that could be incorrectly identified as ships.

The performance of the contrast stretch filter, which maps the minimum intensity in the image to the minimum value in the image range, and the maximum intensity to the maximum value in the range(Figure 4), was also tested and analyzed. This filter was chosen for its effectiveness as a point operator, supporting the independent labeling of each pixel, which enables it to overcome the noise mentioned above by normalizing each pixel's value and increasing the contrast of the image's gray shades. Since the contrast stretch filter parameters are usually not tailored to the values and distribution of the image gray-shades, this filter can be considered to be inferior to histogram equalization. Our solution to this challenge was to use the lowest or highest approximate value of the pixels when implementing the filter using the following equation [19]:

$$P_{out} = \left\lfloor (P_{in} - c) \cdot \left( \frac{255}{d - c} \right) \right\rfloor \tag{2}$$

where $P_{out}$ is the updated value of a pixel, $P_{in}$ is the original value, and $c$ and $d$ are the 2nd and 98th percentile in the image histogram. In addition, values below 0 are rounded up to 0 and values above 255 are rounded down to 255.

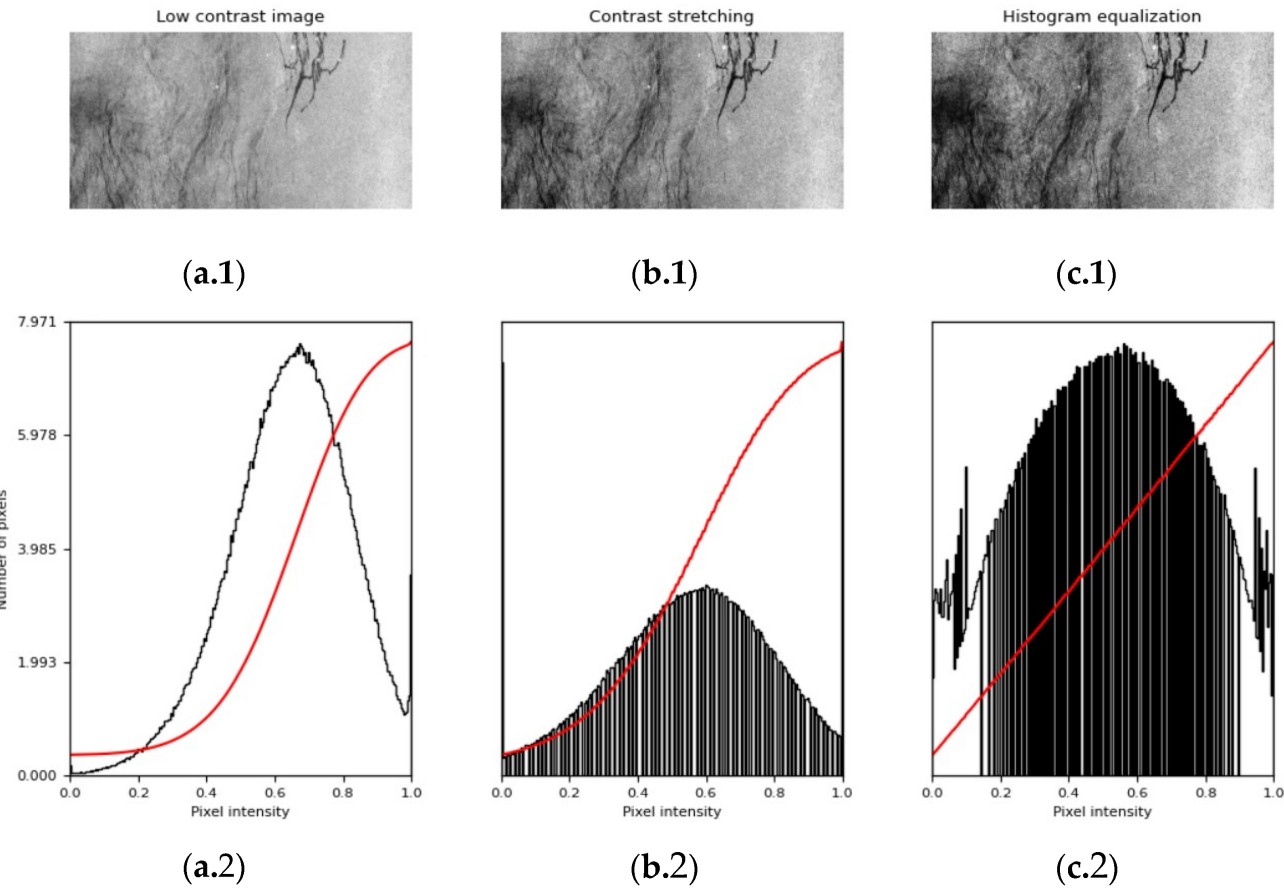

**Figure 4.** Filter comparison: (**a.1**) original image; (**b.1**) image with contrast stretching; (**c.1**) image with histogram equalization. the second row contains image histograms (black) and the cumulative distribution function (red) of each case: (**a.2**) original image; (**b.2**) image with contrast stretching; (**c.2**) image with histogram equalization. As can be seen, each filter affects the image histogram distribution differently.

### 2.4. Model Ensemble

A noticeable difference between the scores of the models on the training set and the scores for the test set was observed, indicating that the neural networks experienced a high generalization error. As undertaken by Zhou [20], it was decided to combine several models in an effort to decrease the generalization error and boost the overall performance. A separate model was created for averaging the output SoftMax score of several models, similar to Simonian and Zisserman [21]. This resulted in a probability distribution over the predicted output classes, where the prediction for each pixel is the class that achieved the highest probability. The ensemble consisted of all six models that were trained: Unet with no filter added, DeepLabv3+ with no filter added, Unet with contrast stretch filter, DeepLabv3+ with contrast stretch filter, Unet with histogram equalization filter, and DeepLabv3+ with histogram equalization filter.

### 3. Results and Discussion

#### 3.1. Experimental Methods

The models were trained and tested on the Oil Spill Detection Dataset using a PyTorch framework and used the Segmentation Model Repository for the implementations of the CNNs. Both the Unet and DeepLabv3+ architectures were trained on images of size $384 \times 384$, which were randomly cropped from each image at every epoch. A total of 100 images was held from the training dataset to form a validation dataset that was later used to examine the optimal learning rate, optimizer, and batch size. The values obtained

were $5 \times 10^{-5}$ for the learning rate and 32 for the batch size, and the selected optimizer was Adam [22]. After choosing these parameters, we combined the validation set and training set into one large training set.

All models were able to reach convergence by epoch 491, but we trained the models for a total of 600 epochs each so we could compare them to the baseline paper [10]. For obtaining the best results, a method of early stopping was used as the model saved was the one that achieved the highest mean IoU score on the test set. Table 1 shows the epoch at which each model obtained its optimal result.

**Table 1.** The required epoch number for each model that obtains its highest mean IoU score.

| Model | Epoch Number with Highest Result |
|---|---|
| Unet no filter | 431 |
| DeepLabv3+ no filter | 383 |
| Unet with contrast stretch | 333 |
| DeepLabv3+ with contrast stretch | 491 |
| Unet with histogram equalization | 265 |
| DeepLabv3+ with histogram equalization | 366 |

Loss Function and Accuracy Metric

The mean intersection over union (IoU) was used as the loss function. The IoU score is a metric that evaluates the resemblance between the prediction and the ground truth; a score of 1 indicates total overlap, while a score of 0 indicates no overlap. For each class, the IoU is defined as:

$$\text{IoU Score} = \frac{|\text{Prediction} \cap \text{Ground Truth}|}{|\text{Prediction} \cup \text{Ground Truth}|} \tag{3}$$

The mean IoU score is the average of all IoU scores, giving each class equal weight in the score and counteracting the effect of class imbalance in the dataset. The loss is defined as follows:

$$\text{IoU Loss} = 1 - \text{mean IoU} \tag{4}$$

Note that an IoU loss of 0 gives complete overlap for the predictions of all classes with their respective ground truth, whereas a loss of 1 means no overlap at all.

*3.2. Final Results*

After training the models for the given number of epochs and saving the models that achieved the best results on the test set, the IoU score was calculated for each class, along with their mean IoU scores. These scores appear in Table 2, along with results from the baseline paper for the Unet and DeepLabv3+ architectures [10].

**Table 2.** Comparison of different filters on the segmentation results of the Unet and DeepLabv3+ in terms of IoU.

| Model | Ocean | Oil Spill | Oil Look-Alike | Ship | Land | Mean |
|---|---|---|---|---|---|---|
| Unet baseline [10] (p. 12) | 93.90 | 53.79 | 39.55 | 44.93 | 92.68 | 64.97 |
| DeepLabv3+ baseline [10] (p. 12) | 96.43 | 53.38 | 55.40 | 27.63 | 92.44 | 65.06 |
| Unet no filter | 95.59 | 51.63 | 47.73 | **50.59** | 96.33 | 68.30 |
| DeepLabv3+ no filter | 95.85 | 49.00 | 51.62 | 39.39 | 93.99 | 65.98 |
| Unet with contrast stretch | 96.21 | 54.2 | 53.42 | 46.25 | 95.8 | 69.18 |
| DeepLabv3+ with contrast stretch | 96.07 | 54.5 | 54.51 | 42.46 | 95.94 | 68.68 |
| Unet with histogram equalization | 96.43 | 51.8 | 55.14 | 46.26 | 96.28 | 69.18 |
| DeepLabv3+ with histogram equalization | 96.35 | 53.78 | 57.70 | 41.37 | 92.3 | 68.3 |
| Ensemble model | **96.78** | **56.1** | **58.88** | 47.28 | **96.59** | **71.12** |

As shown in Figures 5 and 6, applying the histogram equalization filter along with the contrast stretch filter improves the overall performance of both the Unet architecture and the DeepLabv3+ architecture when compared to the baseline results. The results in Table 2 below further imply that filters improve the performance on the Ocean, Oil Spill, and Oil Look-Alike classes, but have a mixed effect on the Ship and Land classes. Averaging the output of the given models for each class, which is presented as the ensemble model, shows further improvement in the results for all classes, except the Ship class.

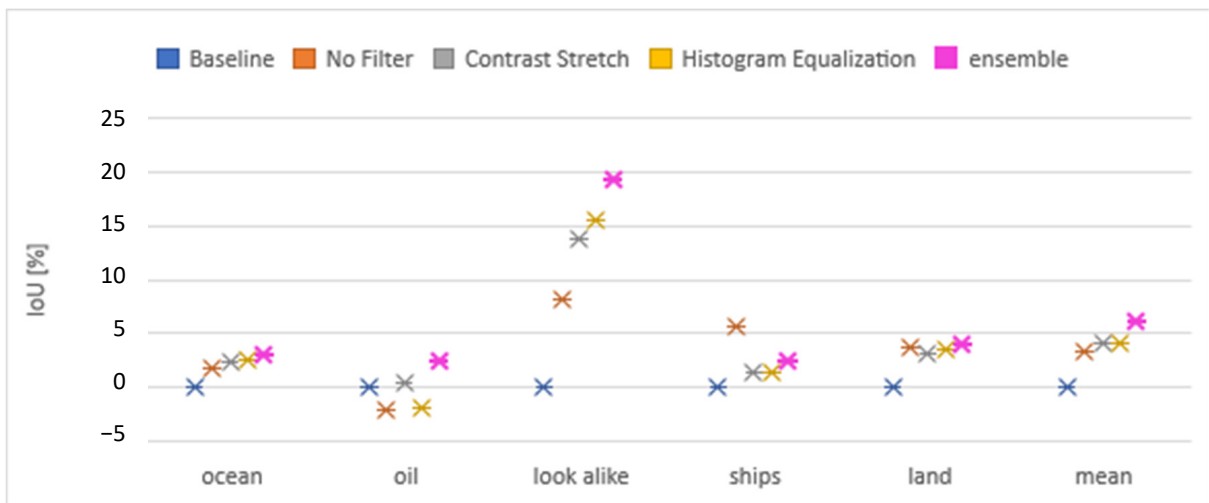

**Figure 5.** Improvement compared to baseline—Unet.

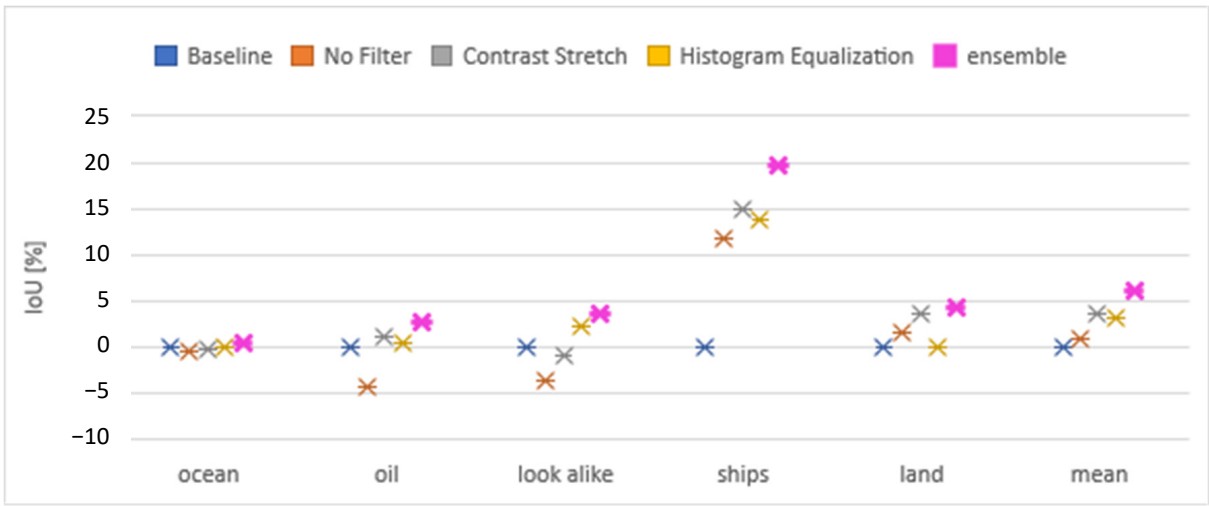

**Figure 6.** Improvement compared to baseline—DeepLabv3+.

The ensemble model achieved a mean IoU score of 71.12%, showing an improvement of ~9.3% compared to the highest score of 65.06% in the baseline research. The ensemble model achieved the following improvements: ~0.3% for the Ocean class, ~4.2% for the Oil Spill class, ~6.2% for the Oil Look-Alike class, 5.2% for the Ship class, and 10% for the Land class, all as compared to the highest performing model in the respective class, as shown in Figure 7. In the baseline research, the highest scores were achieved for either the Unet model or the DeepLabv3+ model, except for the Land class, which achieved its highest score of 63.97% with the LinkNet model.

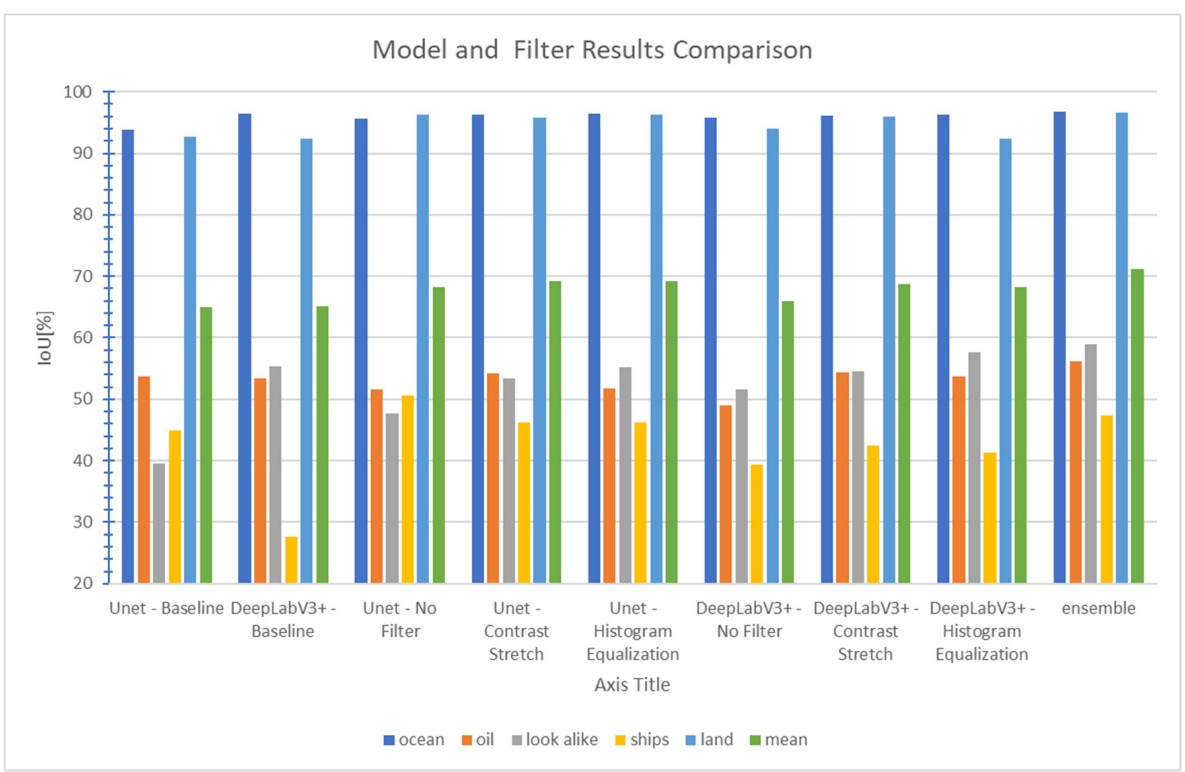

**Figure 7.** Comparison of model and filter final results.

The Ocean class showed an improvement in the IoU score for both filters when applied to either model. The highest score was 96.43% for the Unet model using the histogram equalization filter, which is equal to the score performed by the DeepLabv3+ model in the baseline research. The Oil Spill class, which is our most challenging and significant class, showed an improvement in its detection for both the contrast stretch filter and histogram equalization filters compared to the models without a filter. Using the contrast stretch filter, the models showed an improvement compared to their baseline: a ~1% increase for the Unet architecture and a 2% increase for the DeepLabv3+ architecture. The highest IoU score of 54.5% was achieved by the DeepLabv3+ model with the contrast stretch filter. The Oil Look-Alike class also showed an improvement for both filters compared to no filter. The histogram equalization filter achieved the best results for Oil Look-Alike, showing an increase of ~40% for the Unet architecture and ~4% for the DeepLabv3+ architecture. Regarding the effects of the filters on the Ship and Land classes, there was no conclusive change that could be inferred.

### 3.2.1. Evaluation during Training

The model was evaluated on the test set, by calculating the IoU scores for each class after every epoch of the training, along with the mean IoU score. Figures 8 and 9 show the results for each model.

As shown in Figures 8 and 9, throughout the training, the Ocean and Oil Look-Alike scores were higher when filters were applied, as compared to no filters. This effect can also be seen for the Oil Spill class, although it is more obvious for DeepLabv3+ than for Unet. This reinforces further the results seen in Table 2, which showed a clear improvement when using both filters. Both Unet and DeepLabv3+ regularly reached an IoU score of 0.55 for the Oil Spill class when we applied the histogram equalization filter. However, in the best-performing model, the IoU score of this class was slightly less for both architectures. This indicates that although the results in Table 2 showed better performance for the contrast stretch filter in detecting Oil Spill pixels, both filters are equally beneficial in this regard.

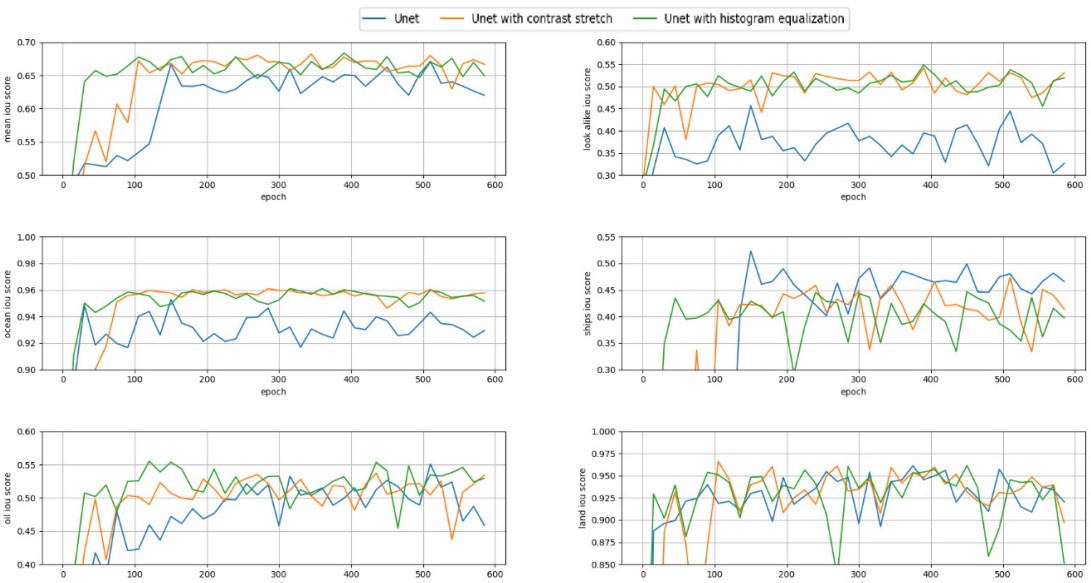

**Figure 8.** Test set IoU scores of the models utilizing the Unet architecture, measured every 25 epochs.

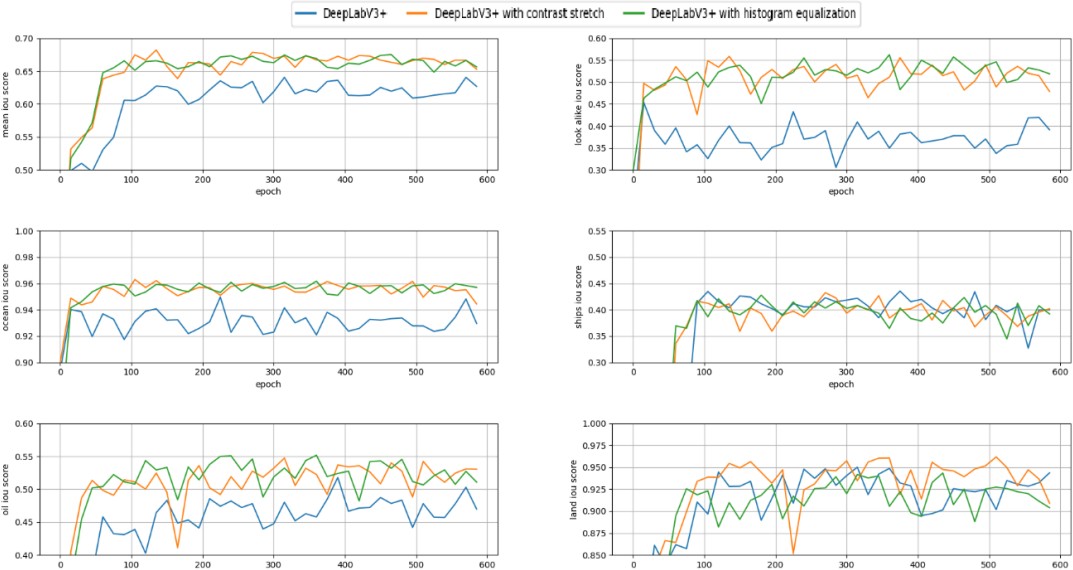

**Figure 9.** Test set IoU scores of the models utilizing the DeepLabv3+ architecture, measured every 25 epochs.

### 3.2.2. Qualitative Results

A visual comparison is shown in Figures 10–12. Figure 10 shows an improvement in the model's ability to correctly predict more Ocean pixels. As a result, the Oil Spill on the right, along with a Ship, could be successfully detected by the models trained on the filtered images. Figure 11 shows that the effects of the filters made the Oil Look-Alike pixels less similar to an Oil Spill, so the models were able to correctly detect the dark pixels as look-alikes. Figure 12 shows three effects of the filters on a noisy image. In the first effect, more Ocean pixels were falsely predicted as Oil Look-Alike on the filtered images. The second effect showed fewer false predictions for the Oil Spill class, as they were mostly replaced by predictions for the Oil Look-Alike class. The third effect was that some correct predictions of the Oil Spill class were replaced by false predictions for the Oil Look-Alike class. This was more clearly seen from the models that applied the histogram equalization filter. The advantage of the ensemble model can be observed in this figure as it was able to eliminate most of the errors produced by the filters while maintaining the correct predictions.

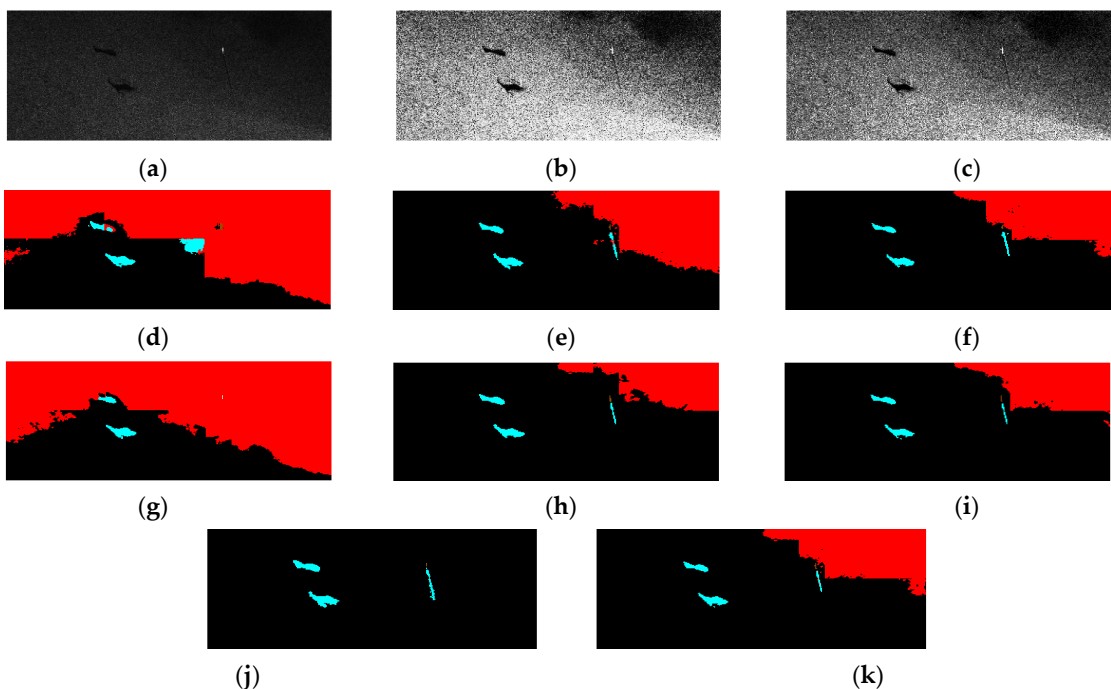

**Figure 10.** Example 1 of the different predictions created by the models: (**a**) original image; (**b**) image with histogram equalization; (**c**) image with contrast stretch; (**d**) DeepLabv3+ prediction; (**e**) DeepLabv3+ with histogram equalization prediction; (**f**) DeepLabv3+ with contrast stretch prediction; (**g**) Unet prediction; (**h**) Unet with histogram equalization prediction; (**i**) Unet with contrast stretch prediction; (**j**) ground truth; (**k**) ensemble model prediction.

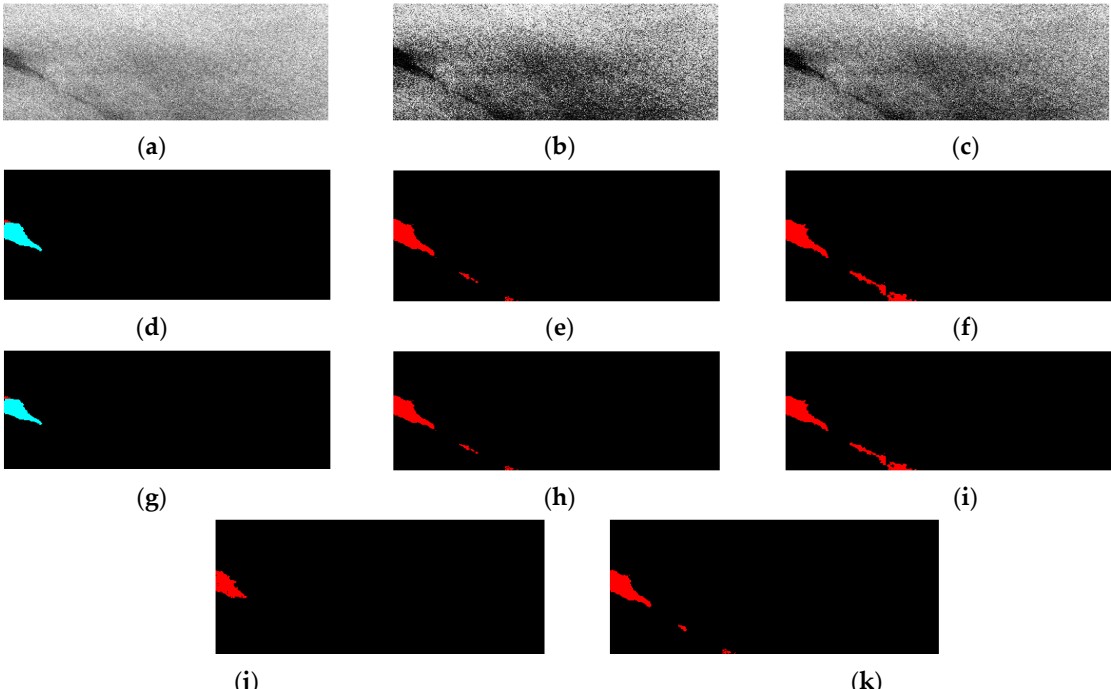

**Figure 11.** Example 2 of the different predictions created by the models: (**a**) original image; (**b**) image with histogram equalization; (**c**) image with contrast stretch; (**d**) DeepLabv3+ prediction; (**e**) DeepLabv3+ with histogram equalization prediction; (**f**) DeepLabv3+ with contrast stretch prediction; (**g**) Unet prediction; (**h**) Unet with histogram equalization prediction; (**i**) Unet with contrast stretch prediction; (**j**) ground truth; (**k**) ensemble model prediction.

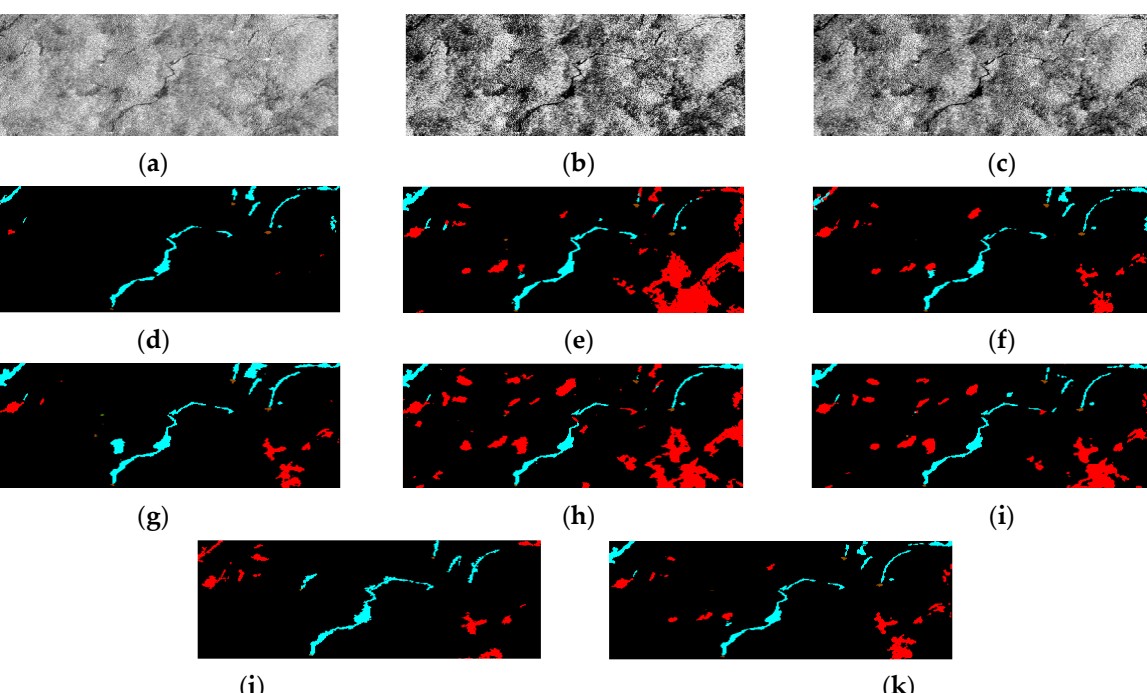

**Figure 12.** Example 3 of the different predictions created by the models: (**a**) original image; (**b**) image with histogram equalization; (**c**) image with contrast stretch; (**d**) DeepLabv3+ prediction; (**e**) DeepLabv3+ with histogram equalization prediction; (**f**) DeepLabv3+ with contrast stretch prediction; (**g**) Unet prediction; (**h**) Unet with histogram equalization prediction; (**i**) Unet with contrast stretch prediction; (**j**) ground truth; (**k**) ensemble model prediction.

## 4. Conclusions

SAR images from satellite monitoring of marine areas and coastlines may be helpful for protecting the ocean environment from oil spills. Analyzing these images to detect oil spills using CNNs with semantic segmentation has shown promising results, but still leaves room for improvement. By applying different data augmentation and image filtering methods, we were able to improve the performance of oil spill detection performed using CNNs in both the Unet and DeepLabv3+ architectures.

Using various image processing techniques, the goal was to distinguish the differences between the labels Oil Spills and Oil Look-Alikes. Moreover, regarding the challenge of the model's generalization error, we found that a possible solution is using the ensemble model, which has increased accuracy and performance due to combining all the techniques mentioned above. This suggests that integrating various aspects and characteristics of SAR images' features can improve the overall detection of marine elements.

Regardless of the architecture chosen, the histogram equalization and contrast stretching filters helped in the detection of the Ocean, Oil Spill, and Oil Look-Alike classes but showed nothing conclusive for Land and Ship classes. Most previous studies in this field did not attempt methods to highlight the textures and outline of the images' elements, i.e., [10,13]. The authors of [14] examined the effect of tested threshold segmentation techniques on the results, which resulted in the loss of information from the original images. The image pre-processing techniques used in this study were able to highlight the differences between elements without losing significant information from the images. This indicates that emphasizing the disparities between the elements of the images, without causing much loss of information, assists in the detection of marine oil spills.

Using an ensemble approach improved the results of all classes and reached the highest mean IoU score of 71.12, showing an improvement of 9.3% over the state-of-the-art DeepLbav3+ model presented in [10], which achieved a mean IoU score of 65.06.

In general, applying these methods has improved the overall mean IoU rate and the IoU score for all the classes examined: Ocean, Land, Oil Spill, Oil Look-Alike, Ship.

In conclusion, the following chart (Figure 13) represents the steps suggested in the Oil Spill detection sequence as a solution system for this challenge:

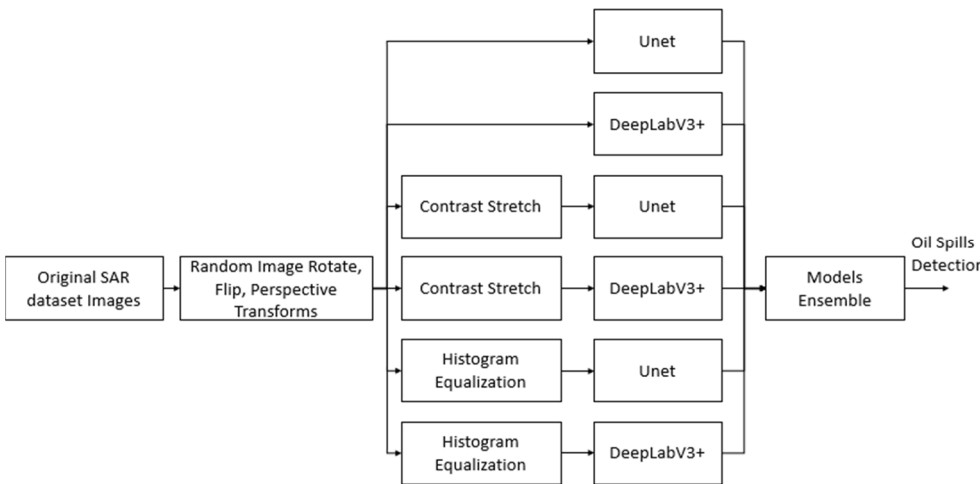

**Figure 13.** Oil Spill detection system—suggested sequence.

*Future Work—Dual 'Oil Look-Alike' Relabeling*

Although we achieved significant improvements in this work, there is room for further improvement. We plan to apply different methods of image processing to a wider variety of algorithms. Specifically, further highlighting the differences between images' features may improve the label detection performance. In addition, relabeling may help to further separate the Oil Spill and Oil Look-Alike pixels. To date, recent studies have considered only five labels: Ocean, Oil Spill, Oil Look-Alike, Ship, and Land. However, defining all darker spots in the ocean as Oil Spill or as Oil Look-Alike is too general. Many Oil Look-Alike spots are differed by several characteristics. Splitting Oil Look-Alike into sub-labels can highlight the differences between them and improve the overall Oil Look-Alike detection.

For future work, we propose using two types of Oil Look-Alike labels, based on their dark spots and other characteristics: (1) feathery, tail, and angular winding shapes ('Sharp Look-Alike'); and (2) patch and droplet spots ('Patch Look-Alike'). Having two types of Oil Look-Alikes can help us distinguish them from the oil spills in the training phase and improve the accuracy of automatic oil spill detection.

Figures 14–16 present the original satellite image along with the current labeling, with a single Oil Look-Alike, and the new labeling, with dual Oil Look-Alike types.

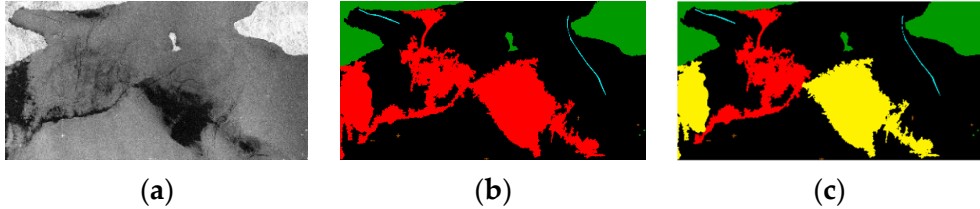

**Figure 14.** (**a**) Satellite image; (**b**) single 'Oil Look-Alike' type (red); (**c**) two 'Oil Look-Alike' types (red as Sharp and yellow as Patch Look-Alike).

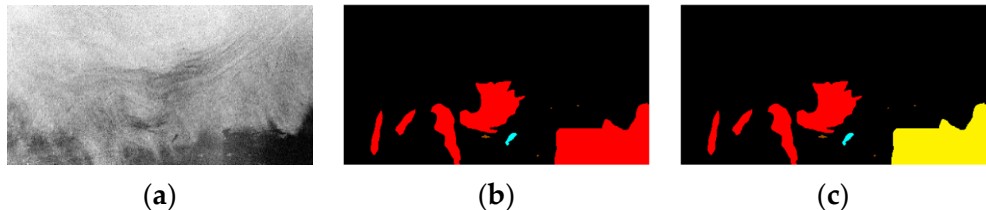

**Figure 15.** (**a**) Satellite image; (**b**) single 'Oil Look-Alike' type (red); (**c**) two 'Oil Look-Alike' types (red as Sharp and yellow as Patch Look-Alike).

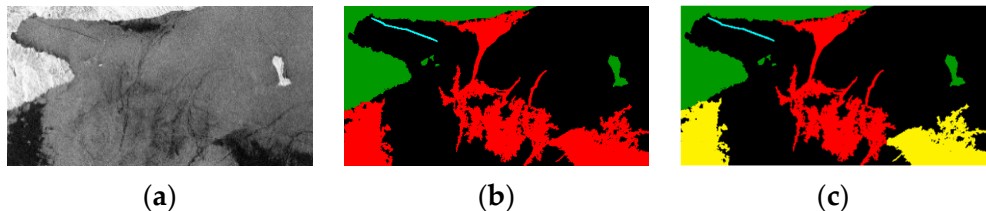

**Figure 16.** (**a**) Satellite image; (**b**) single 'Oil Look-Alike' type (red); (**c**) two 'Oil Look-Alike' types (red as Sharp and yellow as Patch Look-Alike).

**Author Contributions:** Conceptualization—A.S.; writing—original draft, R.R., N.K., Y.G., G.S. and E.K.; writing—review and editing, A.S. and E.K.; analysis—R.R., N.K. and Y.G.; all authors reviewed the manuscript. All authors have read and agreed to the published version of the manuscript.

**Funding:** This research was funded by European Union's Horizon 2020 Research and Innovation Program (H2020-BG-12-2016-2), grant number No. 727277—ODYSSEA (Towards an integrated Mediterranean Sea Observing System). The article reflects only authors' view and that the Commission is not responsible for any use that may be made of the information it contains.

**Institutional Review Board Statement:** Not applicable.

**Informed Consent Statement:** Not applicable.

**Data Availability Statement:** The database used in this study is available on request from the authors of [10]. The data presented in this study, besides the database, are available on request from the corresponding author.

**Acknowledgments:** The authors would like to thank the ESA for gathering the SAR images, and the EMSA for providing information on the geographic coordinates and timestamps via the Clean Sea Net service.

**Conflicts of Interest:** The authors declare no conflict of interest.

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
