# Peer review of "Automatic Recognition of Oil Spills Using Neural Networks and Classic Image Processing"

_water, doi:10.3390/w14071127_

Round 1

Reviewer 1 Report

Journal Water (ISSN 2073-4441)

Manuscript ID water-1608566

Title Automatic Recognition of Oil Spills Using Neural Networks and Classic Image Processing

Although the topic and results are interesting, the presentation should be improved. Below please see my comments:

The ‘Abstract’ must be revised and improved, in order to be better correlated with the content of the paper and objectives.  Please consider that the results should represent the most important part of the ‘Abstract’. You should address briefly the obtained results.

  1. What specific improvements consider regarding the methodology in this paper and what does it add to the subject area compared with other published material? The methodology that needs to improve is  general (trained and tested the models on the Oil Spill Detection Dataset using a PyTorch framework and used the Segmentation Model Repository for the implementations of the  CNNs), there is no innovation in the paper
  2. Discussion is expected to be an important part of this paper, focused on comparisons with other approaches and authors. (please improve it )
  3. Please ensure that the figures have a resolution of at least 300dpi
  4. Please follow the MDPI guideline for formatting the references and carefully check the correctness of each and every reference.

Dear authors,

 I would like to thank you for the effort of the article. . Regarding the manuscript, the current form is rejected (research not conducted correctly) revisions. Some aspects must be improved (ex. abstract, Methodology, discussion, and conclusion).

Thank you for your contribution

Best Regards

Reviewer 2 Report

The manuscript of the paper titled "Automatic Recognition of Oil Spills Using Neural Networks and Classic Image Processing" has been carefully reviewed, and the authors apply new algorithmic techniques to improve early identification of marine oil spills. The objective of the dissertation research is important but seems to be more suitable for submission to journals related to remote sensing technology and geographic information. The authors should clearly show the current state of oil spills and possible derivative applications (subsequent processing procedures), and the bias (misjudgment) instances caused by different algorithms should be appropriately discussed, and the limitations of these methods should be presented to the future work (section 5). From my personal point of view, the lack of verified cases and the lack of innovation in the image recognition algorithm have led to insufficient discussion of the limitations of the method, and should be submitted to other journals.

Reviewer 3 Report

I found the paper interesting and I have enjoyed reading tit. I have few comments on the paper, as follows:

  • the abstract is too long - please try to be more concise.
  • please discuss more in depth the selected references from the field
  • please add explanations to the variables introduced in equation (1) and eventually a reference (if needed)
  • please increase the readability of the figures - especially in the case of figure 1 as the text below the figures is hard to read - you can choose to increase the font or to remove the text as the title of the figure contains the explanation
  • please add explanation to the variables is equation (2)
  • please insert the caption of the tables above the tables

Last, I found section 5 interesting and even though I think figures should not be placed in the final sections of the paper, in this case I believe that they present more clearly the future work the authors are envisioning.

Reviewer 4 Report

The article addresses an important topic, which does have an effect on environmental, engineering and other important fields of the water sector. However, as it stands seems to be an article that is too focused on image treatment and the improvement image detection techniques that are here applied to oil spill but that could be applicable to any other similar detection problem based on  images or satellite similar data. Therefore, I feel inclined to suggest a submission to a journal with higher expertise and audience on this topic. Nevertheless assuming that it could be fitted into Water (the journal), I believe the following aspects need to be attended:

  • The results are too descriptive and the comparison with similar cases and other relevant research in the literature is scarce and needs to be improved before being considered for publication.
  • Also the section related to future works, should be somehow embedded in the conclusions or throughout the results discussion, as it stands it looks like a Master or PhD thesis organization...
  • Additionally, the data set presented enables the validation of the approach and so on. But would be nice to have some additional (if possible) validation by means of comparison with other cases reported in the literature.
  • Finally, it is a bit unclear where the novelty of the methodology is. It looks a bit like a compilation of a set of methods more than an actual novel approach that improves the state of the art per se.

Below some minor points for recommendation:

I believe that along with refs [1] and [2] it would be important to stand out the numerous losses related to oil disasters, oil spills and how these tend to shape the offshore industry, naval activities and so on towards the improvement of regulations and norms to avoid these events. A set of disasters related to this has been compiled in DOI: 10.1680/jmaen.2019.172.4.118 

L78 to 79 - how did the authors define the proportion between the images training data set and the test data set?

L109 - authors say the new method compares positively with state of the art and measures of good performance have been improved. But they only refer to work [8]. Are they specifically stating they outperformed the scores in [8] or are they also referring to other works in the literature, for which 9.3% of improvement is a worthy achievement

Is it possible to enlarge Figure 1 for a better view?

A lot of figures have a title which is not needed, as the figure is described in the caption.

Round 2

Reviewer 2 Report

Many thanks to the authors for their efforts in responding to the reviewers' comments. However, with regard to the application of image analysis, these improvements and potential limitations of different approaches should be addressed. Therefore, I do not recommend that the current manuscript be published in Waters.

Reviewer 4 Report

The paper has improved and can now be published.

Congratulations

This manuscript is a resubmission of an earlier submission. The following is a list of the peer review reports and author responses from that submission.